# Transgenic Ectopic Overexpression of Broad Complex (*BrC-Z2*) in the Silk Gland Inhibits the Expression of Silk Fibroin Genes of *Bombyx mori*

**DOI:** 10.3390/insects11060374

**Published:** 2020-06-16

**Authors:** Jiangshan Cong, Cuicui Tao, Xuan Zhang, Hui Zhang, Tingcai Cheng, Chun Liu

**Affiliations:** State Key Laboratory of Silkworm Genome Biology, Southwest University, Chongqing 400716, China; shan0164@163.com (J.C.); tcc206783@email.swu.edu.cn (C.T.); zxakawdttj@163.com (X.Z.); zh1257554715@163.com (H.Z.); chengtc@swu.edu.cn (T.C.)

**Keywords:** broad complex, fibroin genes, gene regulation, silk proteins, *Bombyx mori*

## Abstract

*Bombyx mori* silk protein genes are strictly turned on and off in different developmental stages under the hormone periodically change. The broad complex (*BrC*) is a transcription factor mediating 20-hydroxyecdysone action, which plays important roles during metamorphosis. Here, we observed that two isoforms of *BmBrC* (*BmBrC-Z2* and *BmBrC-Z4*) exhibited contrasting expression patterns with fibroin genes (*FibH*, *FibL* and *P25*) in the posterior silk gland (PSG), suggesting that BmBrC may negatively regulate fibroin genes. Transgenic lines were constructed to ectopically overexpress *BmBrC-Z2* in the PSG. The silk protein genes in the transgenic line were decreased to almost half of that in the wild type. The silk yield was decreased significantly. In addition, the expression levels of regulatory factors (*BmKr-h1* and *BmDimm*) response to juvenile hormone (JH) signal were inhibited significantly. Then exogenous JH in the *BmBrC-Z2* overexpressed lines can inhibit the expression of *BmBrC-Z2* and activate the expression of silk protein genes and restore the silk yield to the level of the wild type. These results indicated that BmBrC may inhibit fibroin genes by repressing the JH signal pathway, which would assist in deciphering the comprehensive regulation mechanism of silk protein genes.

## 1. Introduction

Insects undergo developmental transitions during their growth, which are aided by two major hormones: 20-hydroxyecdysone (20E) [1] and juvenile hormone (JH) [2]. The hormone 20E is necessary for the transformation during larva–larva and larva–adult, while JH maintains the larval growth process. Both 20E and JH change periodically that resulted in the repression or activation of the target genes in insects [3]. The topic of the relationship of hormones and gene regulation has always been a hotspot research in insect development. The silkworm, *Bombyx mori*, is an important economic insect and a lepidopteran model with applications to biologic science, modern agriculture and biotechnological industry [4]. Their silk-producing organ, the silk gland is considered an ideal system to elucidate the relationship between gene expression and hormone because the silk protein genes are turned on and off during the intermolt to molt stage under JH and 20E control with temporal specific manner [4].

The silk gland of the silkworm is divided into three compartments: the anterior silk gland (ASG), the middle silk gland (MSG) and the posterior silk gland (PSG). As the spatio-specific protein data, the MSG synthesizes sericin proteins, including Ser1, Ser2, Ser3 and Ser4 [5,6,7,8]. The PSG synthesizes fibroin proteins, including fibroin heavy chain (FibH), fibroin light chain (FibL) and fibrohexamerin/P25 [9]. According to the temporal-specific gene expression data, the sericin and fibroin genes are highly expressed during the inter-molting stage, especially in the last instar, but are hardly expressed during the molting stage [10,11]. Currently, the general opinion is that the spatio- and temporal-specificities of silk protein genes are controlled by the signaling pathways of segmental differentiation factors and hormones at the transcriptional level [4].

Silk protein genes are repetitively turned on and off in the inter-molting and molting processes during the whole larval stage [10]. In order to reveal the molecular mechanisms of transcriptional regulation, especially of the fibroin genes, regulatory elements or regions upstream of the promoters of these genes were identified [12,13,14,15,16,17,18,19,20]. Correspondingly, many factors were identified to regulate the transcription of silk protein genes, including BmFkh/SGF-1 [21], SGF-2 [22], POU-M1/SGF-3 [23], SGF-4 [13], FBF-A1 [13], FMBP-1 [17,24], BmFTZ-F1 [25], BmSage [26] and BmDimm [27]. Most of them function as activators of silk protein genes. For example, BmSage and BmDimm are mainly expressed at the inter-molting stage [26,27]. BmSage can interact with the silk gland factor-1 (SGF-1), form a complex after binding to A and B elements on the *FibH* promoter and activate the fibroin synthesis [26]. BmDimm can interact with BmSage to regulate the transcription of *FibH* by binding to the E-box element and be upregulated by Krüppel homolog 1 (BmKr-h1) by the JH-signaling pathway [27]. Both FMBP-1 and BmFTZ-F1 could repress the activity of *FibH* promoter by binding to the upstream elements at the cellular level, suggesting that they may be repressors of the *FibH* gene [24,25].

Broad complex (BrC) is one of the early responding transcription factors of 20E, which has been widely studied as a medium of the 20E and JH-signaling pathways [28,29,30,31]. In *B. mori*, *BrC* gene can produce four isoforms (*Z1–Z4*) with an evolutionarily conserved bric-a-brac–tramtrack–broad (BTB) domain and a zinc–finger DNA-binding domain and four another isoforms (*NZ1–NZ4*) without a zinc finger domain [2,32,33]. Among them, *BmBrC-Z2* and *BmBrC-Z4* perform multiple biologic functions in different tissues and developmental stages. For example, female moths treated with *BmBrC-Z2* dsRNA laid fewer and whiter eggs [34]. BmPOUM2 binds only to BmBrC-Z2 to collaboratively regulate BmVg expression by 20E induction to control vitellogenesis and egg formation [35]. BmBrC-Z4 enhances the expression of chitinase 5 during molting and metamorphosis [36]. During the development of wing disc, BmBrC-Z2 regulates the cuticle protein gene *BmWCP10* induction by 20E-signaling pathway [37], and BmBrC-Z4 regulates the pupal-specific expression of the wing disc cuticle protein gene BmWCP4 through BmPOUM2 [38]. In the innate immune response, BmBrC-Z2 and BmBrC-Z4 also play an important biologic role in regulating the expression of immune-related genes, including lysozyme and lebocin [39,40].

According to our previous transcriptomic analysis of the silk glands from molting to inter-molting stages, *BmBrC-Zs* were significantly overexpressed, but the expression of the fibroin genes at the molting stage was almost absent [11]. Synthetic oligopyrrole carboxamides can downregulate the expression levels of various isoforms of *BmBrC* gene in the silk gland, which resulted in statistically enhanced cocoon weight, shell weight and silk yield [41]. Therefore, we hypothesized that *BmBrC* may be involved in the negative regulation of fibroin genes. To test the hypothesis, in this study, we further investigated their expression patterns from molting to inter-molting stages by RT-PCR. Transgenic lines with ectopic overexpression of *BmBrC-Z2* in the PSG was obtained to investigate the phenotype and gene expression levels. These results may provide an insight into the regulatory mechanism of silk proteins in the silkworm.

## 2. Materials and Methods

### 2.1. Experimental Insects

The silkworm strain Dazao was used for transgenic microinjection was obtained from the gene resource library of domesticated silkworm, Southwest University, China. Silkworm larvae were fed with fresh mulberry leaves at 25 °C with a photoperiod of 12 h light/12 h dark.

### 2.2. Semi-RT-PCR and qRT-PCR

Total RNA was extracted using TRIzol reagent (Invitrogen, USA) according to the protocol provided by the manufacturer. For reverse transcription, we used the PrimeScript RT reagent Kit (Takara, Kyoto, Japan) according to the manufacturer’s instructions. The primers used for semi-RT-PCR are given in Table 1. The following conditions: 94 °C for 30 s, followed by 30 cycles at 94 °C for 10 s, 60 °C for 15 s and 72 °C for 30 s, then elongation at 72 °C for 7 min and held at 16 °C. The ribosomal *protein L3* (*RPL3*) gene of the silkworm was used as the reference gene.

Quantitative RT-PCR (qRT-PCR) was conducted with the ABI7500 real-time PCR machine (Applied Biosystems, Foster, CA, USA) using FastStar Universal SYBR Green Master (Roche, Switzerland). Each qRT-PCR reaction was performed under the following conditions: denaturation at 95 °C for 10 min, followed by 40 cycles at 95 °C for 10 s, 60 °C for 30 s and 72 °C for 35 s. RPL3 gene was used as the reference. Three repeat experiments were set up. The expression of the target gene was calculated by Ct value and graphics were created based on this data. The primers used are shown in Table 1.

### 2.3. Transgenic Overexpression of BmBrC-Z2 in the PSG

A transgenic ectopic overexpression vector was constructed. The sequence encoding BmBrC-Z2 and a MYC-tag were cloned into the PMD-19 vector between the *BamH*I and *Not*I, then sub-cloned into the mid-vector containing a modified *FibH* promotor, which was deleted the predicted Br–C binding sites. Finally, the unit [MFibH-BmBrC-Z2-myc-Sv40] was sub-cloned into the *piggyBac* vector [3xP3-Red-Sv40] with the red fluorescence marker by *Asc*I digestion site. The recombinant vector [3xP3-Red-Sv40-MFibH-BmBrC-Z2-myc-Sv40] was transformed into the *E. coli* strain Trans1-T1 cells (TransGen, Beijing, China), screened for positive clones and the plasmid was extracted. The piggyBac transposase-expressing plasmid pHA3PIG helper was obtained from a previous study [42]. Transgenic plasmid and helper plasmid were mixed both to 500 µg/µL and injected into preblastoderm embryos at 2 h after oviposition. Positive individuals were screened in the pupal stage by red fluorescence using a fluorescence stereomicroscope (Olympus, Tokyo, Japan). The phenotypic characters were investigated between the transgenic overexpressing *BmBrC-Z2* lines and the wild type as per the protocols given in our previous study [43].

### 2.4. Juvenile Hormone Treatment

Overexpressing BmBrC-Z2 lines on the 1st day of the 5th instar larvae (V1stD) were treated with 1 µL JH analogs methoprene (Sigma, St. Louis, Missouri, USA) on the back by the pipette with the concentration of 2 µg/µL. For the controls, equal volumes of DMSO were applied. A quarter of the treatment samples were collected the 2nd day of the 5th instar larvae (V1stD) and stored at −80 °C for subsequent investigation of gene expression patterns. The remaining larvae were raised to the pupal stage and were used to investigate the phenotypic data, including total weight of cocoons and weight of cocoon layer.

### 2.5. Statistical Analysis

All data were statistically analyzed by *t*-test. Each set of data were repeated three times. Asterisks were used to indicate significant differences (* *p* < 0.05; ** *p* < 0.01; *** *p* < 0.001).

## 3. Results

### 3.1. Expression Patterns of BmBrC-Zs and Fibroins in the Silk Gland

Previous studies showed that *BmBrC* was highly expressed in the PSG during the 4th molt (IVM) stage, while fibroin genes were almost undetectable by northern blot and transcriptomic analysis [11,32]. Here, we investigated the expression patterns of three isoforms (*BmBrC-Z1*, *Z2 and Z4*) and fibroin genes (*FibH, FibL and P25*) using semi-RT-PCR (Figure 1A). The results indicated that *BmBrC-Zs* were expressed from the beginning of IVM to 1st day of the 5th instar stage (V1stD). By qRT-PCR analysis, the relative expression level of *BmBrc-Z2* was higher than that of *BmBrc-Z4* (Figure 1B). The expression levels of fibroin genes were gradually inhibited at the IVM and the expression levels of fibroin genes started to recover at V1stD when the expression of *BmBrC-Zs* decreased. Thus, we suspected that *BmBrC* may negatively regulate the expression of fibroin genes.

### 3.2. Transgenic Ectopic Overexpression of BmBrC-Z2 in the PSG

Previous studies on transcription factors regulating the expression of silk protein genes confirmed it in the cellular level [24,25,26,27]. To confirm the biologic function of *BmBrC-Z2* in vivo, a transgenic vector with ectopic overexpression of *BmBrC-Z2* was constructed with a modified *FibH* promotor, in which the predicted Br–C binding site was deleted (Figure 2A). After embryo injection and screening from 366 eggs, fourteen individuals positive for overexpression of *BmBrC-Z2* were obtained. The investigation showed that in the silk glands, there were no obvious phenotypic differences between the transgenic lines and the wild types in the V5thD larvae (Figure 2B). Interestingly, the cocoons of transgenic lines became thinner and lighter color compared with those of the wild types (Figure 2C). The cocoon layer ratios decreased significantly in both the male and female individuals (Figure 2D, n = 50). The results indicated that ectopic overexpression of BmBrC-Z2 in the PSG can lead to a decline in the silk protein synthesis.

### 3.3. Transgenic Overexpression of BmBrC-Z2 Downregulates Gene Expression

To investigate the effect of overexpression of *BmBrC-Z2* in the PSG, genes encoding silk proteins and transcription factors were detected by qRT-PCR at the V5thD (Figure 3). The results showed that *BmBrC-Z2* was increased significantly in the transgenic line than those of the wild type (Figure 3A). All of the silk protein coding genes, including *FibH*, *FibL* and *P25*, were inhibited in the transgenic line (Figure 3B). Interestingly, the expression of two transcription factors, *BmKr-h1* and *BmDimm*, was also significantly inhibited (Figure 3A). We also tested the endogenous *BmBrC-Z4* and found that it was significantly upregulated in the transgenic line (Figure 3A). Our previous study had confirmed that these two transcription factors were important regulators to activate the expression of *FibH* gene by the JH-signaling pathway [27]. These results not only indicated that the ectopic overexpression of BmBrC-Z2 may inhibit the expression of silk protein genes in the PSG of the V5thD larvae, but also showed that the overexpression of BmBrC-Z2 affected the positive regulatory factors of fibroin genes.

### 3.4. Exogenous JH Analog Can Rescue the Phenotype of the Transgenic Line Overexpressing BmBrC-Z2

To further determine whether the overexpression of BmBrC-Z2 repressed the JH-signaling pathway, we treated the transgenic line and wild type larvae at V1stD with exogenous JH analog methoprene (JHA), where dimethyl sulfoxide (DMSO) was used as the control (Figure 4). The results showed that the cocoon phenotypes of the transgenic line overexpressing BmBrC-Z2 were rescued similar to those of the wild type with the application of JH analog (Figure 4A). Further, the cocoon layer ratios reached 8.9% and 11.9% in the female and male individuals, respectively, which were close to that of the wild type (Figure 4B). The expression levels were tested by qRT-PCR in the transgenic line after JH analog treatment. Interestingly, the expression of *BmBrC-Z2* and *BmBrC-Z4* was inhibited (Figure 4C). The expression levels of *BmKr-h1* and *BmDimm* were significantly increased. In addition, the expression levels of all the three fibroin genes were significantly increased (Figure 4D). These results indicated that the exogenous JH analog can reverse the effect of the ectopic overexpression of *BmBrC-Z2* in the PSG to increase the expression levels of silk protein genes.

## 4. Discussion

The transcriptional regulation of silk protein genes has always been a research hotspot in research on silkworms. Silk protein genes are repeatedly turned on and off from the inter-molting to the molting stages and are controlled by a large number of positive and negative transcriptional factors. In this study, we constructed a transgenic line to ectopically overexpress BmBrC-Z2 using the *FibH* promoter in the PSG at the inter-molting stage. The phenotype and molecular investigation demonstrated that the ectopic overexpression of BmBrC-Z2 not only inhibited the expression of silk protein genes and silk protein synthesis, but also repressed the expression of positive regulatory factors BmKr-h1 and BmDimm, which respond to the JH signal to activate the expression of silk protein genes [27].

Several transcription factors have been reported to inhibit the expression of silk protein genes at the molting stage. Both of FTZ-F1 and FMBP-1 could downregulate the expression of *FibH* at the cellular level by binding to the regulatory element on its promoter [24,25]. An element −397 to −389 upstream of the *FibH* promoter was bound by FTZ-F1 to inhibit the promoter activity [25]. FMBP-1 repressed *FibH* promoter activity by directly binding to the −152 to −125 element in the *FibH* promoter region [24]. However, evidence on their in vivo regulatory functions is still mostly absent. Among most of the previous studies, the activity of *FibH* promoter was very weak in the silkworm cell lines. For the activators of *FibH* gene, such as SGF1, Sage and Dimm, these results may be credible [26,27], but for inhibitors, the cellular-level results may be insufficient to explain the molecular regulatory mechanisms in vivo, as the silk gland is a specialized organ. In this study, to explore whether BmBrC-Z2 could inhibit the expression of silk protein genes in vivo, transgenic lines were obtained by the ectopic overexpression of BmBrC-Z2 using a modified *FibH* promotor in the PSG (Figure 2). In the overexpression–transgenic line, the relative mRNA level of *BmBrC-Z2* was significantly increased in the PSG at the fifth instar stage (Figure 3), the expression levels of silk protein genes were also significantly inhibited to almost half of the wild type, thus resulting in a significant reduction in the cocoon weight and cocoon layer ratio. We believed that *BmBrC* was a strong inhibitor of silk protein genes, but the high efficiency of inhibition is still unknown which need further explore in our next step.

*FibH* promoter was very specific and strong to promote foreign gene expression in the PSG [42]. Although the promoter was modified by deleting the BmBrC-Z2 binding site to eliminate its influence, the expression level of *BmBrC-Z2* in the transgenic line was not as high as we expected. It should be noted that the overexpression of BmBrC-Z2 significantly downregulated the expression of *BmKr-h1* and *BmDimm* in the PSG at the inter-molting stage (Figure 3). Thus, we assumed that the reduction of *BmKr-h1* and *BmDimm* may be the reason behind the low activity of the modified *FibH* promoter in the overexpression of *BmBrC-Z2* in the transgenic line. JH-mediated *BmKr-h1* repressed the expression of *BmBrC* by binding to the *BmBrC* promoter during the larval-adult metamorphosis [31,44]. Our previous research reported that BmKr-h1-BmDimm cascade response to the JH signal could activate the *FibH* gene expression [27]. After JH analog treatment of the *BmBrC-Z2* overexpression–transgenic line, the expression levels of *BmKr-h1* and *BmDimm* were significantly increased, and then the expression levels of *BmBrC-Z2* were inhibited significantly. Moreover, the expression levels of all the three fibroin genes were significantly increased (Figure 4). It is suggested that JH-signaling was inhibited when ectopically overexpressing *BmBrC-Z2* in the PSG at the inter-molting stage. However, how BmBrC-Z2 repressed the expression of *BmKr-h1* and *BmDimm* needs to be explored.

In addition, the endogenous *BmBrC-Z4* was significantly upregulated in the inter-molting stage and was also downregulated when treatment of JH in the transgenic line (Figure 3A and Figure 4C), suggesting that overexpression of *BmBrC-Z2* may function as a feedback regulation on ecdysone-signaling pathway. This present study is the inadequacy of the lack of the in vitro activity analysis of promoters of silk protein genes and genetic loss-of-function of *BrC* in the silk gland in vivo. The promoters of silk protein genes are almost hardly active in silkworm cell lines, which is not applicable to the negative regulation of BrC to the silk protein genes in vitro. In addition, knocking-out *BrC* may not be suitable because BrC is very crucial in the hormone-signaling pathway for the development and physiology in the silkworm life cycle [28,29,30]. Based on the present and previous studies [11,27,31], here, we propose a hypothesis for the regulatory mechanism for the inhibition of the expression of silk protein genes by *BmBrC* at the molting stage (Figure 5). BmBrC could inhibit the expression of fibroin genes directly by binding to the upstream regulatory elements or indirectly by BmKr-h1 and BmDimm cascade in the JH signal pathway. At the inter-molting stage, JH binds to a heterodimer of BmMet2/SRC, which directly activates the expression of BmKr-h1. BmKr-h1 inhibits the expression of BmBrC-Z2 and meanwhile activates the expression of BmDimm and then activates the expression of the fibroin genes in the silkworm. Therefore, our findings will assist in deciphering a comprehensive regulation mechanism of silk proteins in the silkworm.

## 5. Conclusions

The expression regulation of silk protein genes are comprehensive, which repetitively turned on and off in the inter-molting and molting processes. In this study, we found that BmBrC could significantly repress the expression of silk protein genes and several transcriptional regulatory factors by a transgenic ectopic overexpression of BmBrC-Z2 in the inter-molting PSG. It is suggested that at the molting stage *BmBrC* could inhibit the expression of silk protein genes directly by binding to the upstream regulatory elements or indirectly by BmKr-h1 and BmDimm in the JH signal pathway, however, at the inter-molting stage, BmKr-h1 could inhibit the expression of *BmBrC-Z2* and then activate the expression of silk protein genes. Our findings will assist in deciphering a comprehensive regulatory mechanism of silk protein genes by the alternate signals of 20E and JH in the silkworm.

## Figures and Tables

**Figure 1 insects-11-00374-f001:**
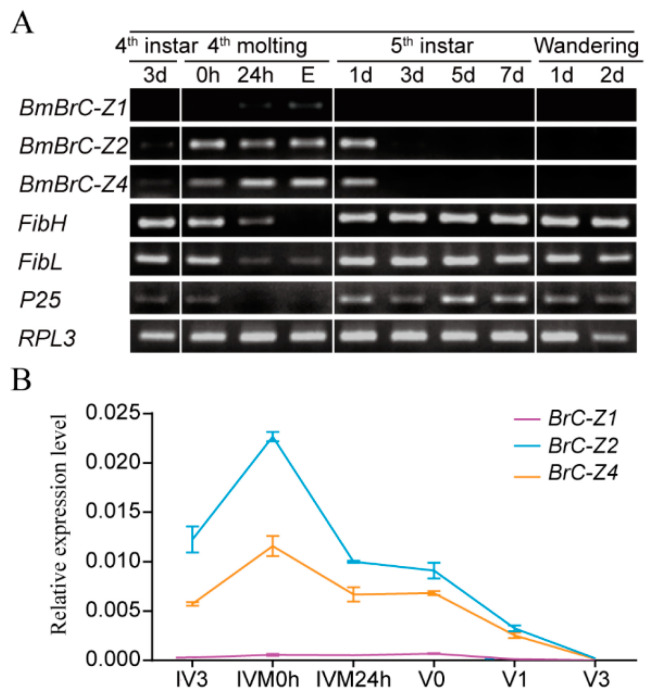
Expression patterns of *BmBrC-Zs* and fibroin genes in *Bombyx mori.* (**A**). Expression levels of *BmBrC-Zs* and *fibroins* in the PSG from 3rd day of the 4th instar to the 2nd day of the wandering stage by semi-RT-PCR analysis; (**B**) relative expression level of *BmBrC-Zs* in PSG at 3rd day of the 4th instar (IV3) to the 3rd day of the 5th instar (V3) by qRT-PCR analysis. *RPL3* shown as a control. d—day; h—hour; E—ecdysis; IV—4th instar; M—molting; V—5th instar.

**Figure 2 insects-11-00374-f002:**
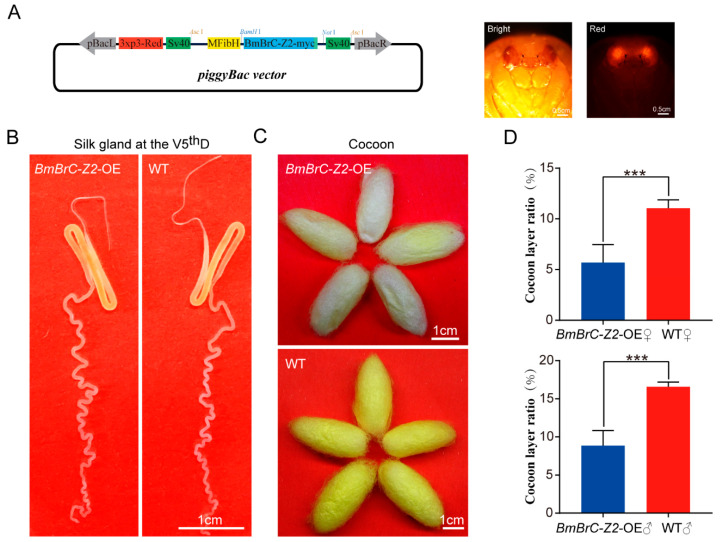
Transgenic overexpression of *BmBrC-Z2* in the PSG. (**A**) Transgenic overexpression vector was constructed to overexpress *BmBrC-Z2* in *B. mori* PSG. The positive transgenic phenotype was observed in the pupae; (**B**) silk gland phenotype exhibited no obvious changes at the V5thD; (**C**) cocoons became thinner and lighter and (**D**) cocoon layer ratio (weight of cocoon layer/total weight of cocoon) was obtained by statistical analysis (n = 50). Wild type was used as the control. WT—wild type; OE—overexpression; ***—*p* < 0.001.

**Figure 3 insects-11-00374-f003:**
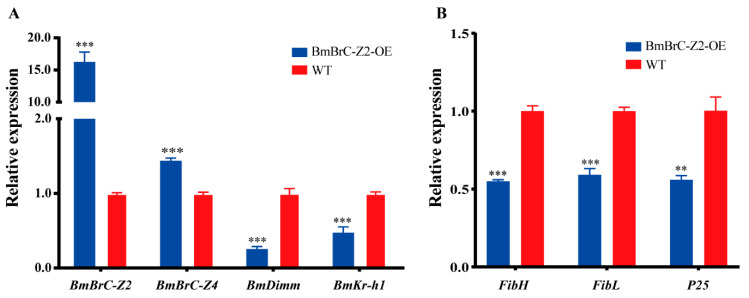
Relative expression levels of *BmBrC-Z2* and related genes in the transgenic line. Including regulatory factors (**A**) *BmBrC-Z2*, *BmBrC-Z4*, *BmDimm*, *BmKr-h1* and (**B**) fibroin genes *FibH*, *FibL*, *P25*. *RPL3* used as the control. **—*p* < 0.01, ***—*p* < 0.001); WT—wild-type; OE—overexpression.

**Figure 4 insects-11-00374-f004:**
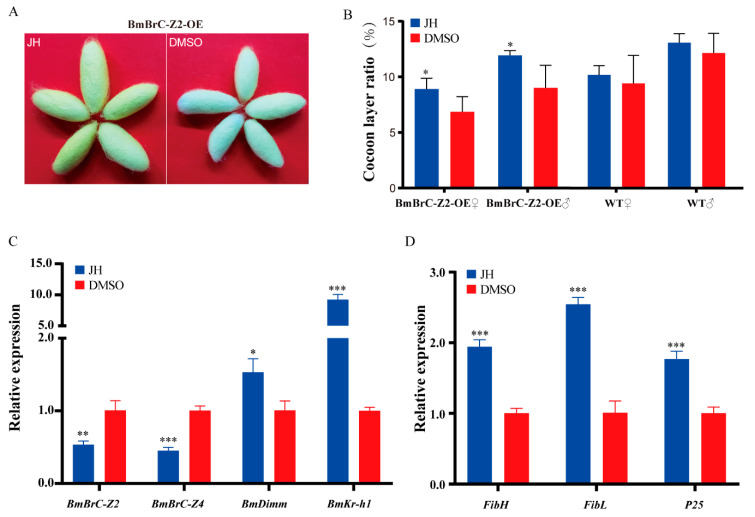
Juvenile hormone (JH) analog rescues the cocoon phenotype of the transgenic overexpression line. (**A**,**B**) Phenotype observation and cocoon layer ratio statistics (n = 10). Treatments of DMSO was used as the control; (**C**,**D**) relative mRNA levels of genes were investigated by qRT-PCR after treatment of JH analog in the transgenic overexpressing *BmBrC-Z2* line. Including: *BmBrC-Z2, BmBrC-Z4, BmDimm*, *BmKr-h1*, *FibH, FibL, P25*. *RPL3* was used as the control. *—*p* < 0.05, **—*p* < 0.01, ***—*p* < 0.001); WT—wild type; OE—overexpression.

**Figure 5 insects-11-00374-f005:**
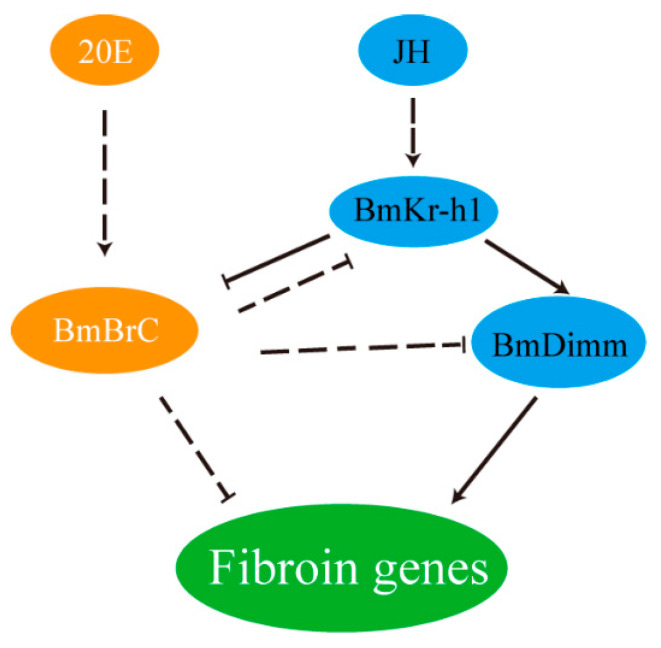
Schematic representation of the hypothesis for molecular regulatory mechanism of BmBrC inhibiting the expression of silk protein genes. At the molting stage, 20-hydroxyecdysone (20E) activates the expression of *BmBrC*. BmBrC-Z2 inhibits the expression of fibroin genes directly by binding to the upstream regulatory elements or indirectly by BmKr-h1 and BmDimm in the JH signal pathway. At the inter-molting stage, JH activates the expression of *BmKr-h1*. BmKr-h1 inhibits the expression of *BmBrC-Z2* and meanwhile activates the expression of *BmDimm*, then activates the transcription of fibroin genes in the silkworm.

**Table 1 insects-11-00374-t001:** Primers used in the semi-RT-PCR and quantitative RT-PCR.

Purpose	Gene	Forward (5′–3′)	Reverse (5′–3′)
RT-PCR	*BmBrC-Z1*	CCCAAGAAGATTACAGATGCG	AGGTGGCTGGTTAGGGTG
RT-PCR	*BmBrC-Z2*	TCGCTGACAAACACGCTG	ATGGTAAGAACGGCGGAC
RT-PCR	*BmBrC-Z4*	GCCACAAGGTCTTCCGCA	AAGAGCCAGCGGAAGGAT
RT-PCR	*FibH*	CAGCATCAGTTCGGTTCC	GACTCGTTACCGTCGGAATC
RT-PCR	*FibL*	ATACCGATTGGTCACATAACAG	GCAGATAGATGGGCGATAA
RT-PCR	*P25*	AGCCGCTGTGGCAGTTTTG	TAGGTGGCGTTGAAGTATGG
RT-PCR	*RPL3*	TCGTCATCGTGGTAAGGTCAA	TTTGTATCCTTTGCCCTTGGT
qRT-PCR	*BmBrC-Z2*	TCGCTGACAAACACGCTG	ATGGTAAGAACGGCGGAC
qRT-PCR	*BmBrC-Z4*	GCCACAAGGTCTTCCGCA	AAGAGCCAGCGGAAGGAT
qRT-PCR	*FibH*	TCTGTGTCATCTGCTTCATCTCG	TATCCAGGACGAAGTAAGAAACAA
qRT-PCR	*FibL*	ATACCGATTGGTCACATAACAG	GCAGATAGATGGGCGATAA
qRT-PCR	*P25*	AGCCGCTGTGGCAGTTTTG	TAGGTGGCGTTGAAGTATGG
qRT-PCR	*BmKr-h1*	CATCGTTTTCAACATTTTGGCGAG	CACATCACTTTACCATCGGCAGC
qRT-PCR	*BmDimm*	CGTGGAACCCGCATTTGTA	AACCTCGGCAATCCAGTCG
qRT-PCR	*RPL3*	TTCGTACTGGCTCTTCTCGT	CAAAGTTGATAGCAATTCCCT

BrC—broad complex; FibH—fibroin heavy chain, FibL—fibroin light chain; Kr-h1—Krüppel homolog1; RPL3—ribosomal protein L3; Bm—*Bombyx mori*.

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
