# Peer review of "Transgenic Ectopic Overexpression of Broad Complex (BrC-Z2) in the Silk Gland Inhibits the Expression of Silk Fibroin Genes of Bombyx mori"

_insects, 2020, doi:10.3390/insects11060374_

Round 1

Reviewer 1 Report

The manuscript by Cong et al. constructed a transgenic silkworm overexpressing the BrC gene in the silkgland and showed that this gene inhibited the expression of silk fibrion genes and reduces the silk yield. The application of juvanile hormone rescues the silk production, indicating the Brc genes may repress the juvanile hormone pathway and lead to the inhibition of fibroin genes. The manuscirpt is generally well written and clear in expressions. However, there are several issues needs to be adressed before acceptance.

Major point:
1. The method section should be described in more detail, for example, provide the transgenic plasmid construction schemes and procedures in more detail, and how the plasmid is injected into the egg and how methoprene was administered to the larva so the readers can fully comprehend the experimental steps.
2. Figure 4: the JH and DMSO treatments were based on transgenic line, what is their treatments on the WT? How does the expressions of those genes change in the WT when treated with JH? Were the cocoons weight further enhanced in the WT with JH treatment? This would be interesting to know.
3. Last paragraph: the authors can provide a graphic view of their hypotheis on the JH signaling pathway as affected by the Brc gene, as schemes will be more easier to comprehend.
4. There are several BrC genes as described by the authors, is there a reason BrC-Z2 is chosen over the other BrC genes?
Other comments: "Methoprene" does not need to be capitalized all the time.

Author Response

#Reviewer 1

The manuscript by Cong et al. constructed a transgenic silkworm overexpressing the BrC gene in the silkgland and showed that this gene inhibited the expression of silk fibrion genes and reduces the silk yield. The application of juvanile hormone rescues the silk production, indicating the Brc genes may repress the juvanile hormone pathway and lead to the inhibition of fibroin genes. The manuscirpt is generally well written and clear in expressions. However, there are several issues needs to be adressed before acceptance.

Major point:

Point 1: The method section should be described in more detail, for example, provide the transgenic plasmid construction schemes and procedures in more detail, and how the plasmid is injected into the egg and how methoprene was administered to the larva so the readers can fully comprehend the experimental steps.

Response 1: Thank you very much for pointing this problem. In the revised manuscript, we add more description about the experimental steps in the materials and methods section.

Point 2: Figure 4: the JH and DMSO treatments were based on transgenic line, what is their treatments on the WT? How does the expressions of those genes change in the WT when treated with JH? Were the cocoons weight further enhanced in the WT with JH treatment? This would be interesting to know.

Response 2: Thanks for your question. There were no significant change of cocoon layer ratio (cocoon weight/total weight) of the WT after JH treatment (Fig. 4B). In our previous study (Zhao et al., JBC, 2015), JH treatment could significantly increase expression levels of genes, such as BmKr-h1, Bmdimm, and Fib-h. Here, we did not investigate the expression patterns of these genes in the WT when treated with JH and only cited the reference in discussion section.

Point 3: Last paragraph: the authors can provide a graphic view of their hypotheis on the JH signaling pathway as affected by the Brc gene, as schemes will be more easier to comprehend.

Response 3: Thanks for your valuable suggestion, we provide a graphic model to illustrate a potential regulatory mechanism.

Point 4: There are several BrC genes as described by the authors, is there a reason BrC-Z2 is chosen over the other BrC genes?

Response 4: Thank you very much for pointing this problem. In Fig. 1, we found that the expression patterns of BmBrc-Z2 and BmBrc-Z4 were similar by RT-PCR analysis. By qRT-PCR analysis, we found the relative expression level of BmBrc-Z2 was higher than that of BmBrc-Z4. The result was shown as Fig. 1B. Thus, we first chose BmBrc-Z2 to ectopically overexpress in the PSG.

Point 5: Other comments:"Methoprene" does not need to be capitalized all the time.

Response 5: Thanks, we have revised it.

Reviewer 2 Report

In this study, the authors put forward the hypothesis that Bombyx mori Broad-Complex (BrC), a transcription factor mediating 20-hydroxyecdysone action, might negatively regulate fibroin genes. To prove this, they contructed the transgenic silkworm overexpressing the BmBrC-Z2 in the posterior silk gland (PSG). The results showed that the silk protein (FibH, FibL and P25) was decreased significantly compared with the wild-type control, and the expression levels of regulatory factors (BmKr-h1 and BmDimm) response to JH signal were also inhibited. Furthermore, exogenous JH in inhibit the expression of BmBrC-Z2 in the over-expressed lines and activate the expression of silk protein genes and restore the silk yield to the normal level. The results is interesting.

However, the conclusion is not enough solid, as no detailed mechanism of this regulation is involved.

(1)More experimental data are required to support the results, such Western Blotting….only semi-RT-PCR analysis is not enough. At least, Western blot must be provided.

(2)To analyze the gen function, usually gene deletion or knock-down is required. Is it possible to delete or knock down the expression of BrC-Z by RNA interference?

(3)How about the mechanism of this regulation ? Could give the signal pathway ?

(4)Poor English writing. Lots of error in grammar and writing. It is necessary to improve it by native English writers.

others points

(1) Figure 1. The ribosomal protein L3 (RPL3) gene of the silkworm was used as the reference gene.

But, there are significant decrease of RPL3 in Wandering 2d compared with other period in the figure. Besides, it is better to provide the ratio of genes of interest to the control RPL3 because the luminance of RPL3 in 2nd Day of wandering seems to differ from other stages.

(2) Figure 2A. The positive transgenic phenotype was observed in the pupae. Could this positive phenotype be maintained in 4th instar, 5th larva? Please offer the relative expression level of BrC-Z2 between transgenic overexpression line and wild-type during the designed time points. *Figure 4. Comparing with BrC-Z2-OE line treatment with JH and DMSO, the author should also supplement data with wild-type treatment with JH or DMSO respectively.

(3) “ 3.3”, appears twice.

(4) Line130: The expression level started to recover at V1stD instead of V3rdD according to figure1.

(5) Line173: It is inadequate to determine the hypothesis mentioned in this line without monitoring JH concentration in transgenic lines.

(6) Line29: “resulted to” should be “resulted in”. “active the target genes in insects” ?

Author Response

In this study, the authors put forward the hypothesis that Bombyx mori Broad-Complex (BrC), a transcription factor mediating 20-hydroxyecdysone action, might negatively regulate fibroin genes. To prove this, they contructed the transgenic silkworm overexpressing the BmBrC-Z2 in the posterior silk gland (PSG). The results showed that the silk protein (FibH, FibL and P25) was decreased significantly compared with the wild-type control, and the expression levels of regulatory factors (BmKr-h1 and BmDimm) response to JH signal were also inhibited. Furthermore, exogenous JH in inhibit the expression of BmBrC-Z2 in the over-expressed lines and activate the expression of silk protein genes and restore the silk yield to the normal level. The results is interesting.

However, the conclusion is not enough solid, as no detailed mechanism of this regulation is involved.

Point 1: More experimental data are required to support the results, such Western Blotting….only semi-RT-PCR analysis is not enough. At least, Western blot must be provided.

Response 1: Thank you very much for your comments. As you said, we also hope to test BrC-Zs in the protein level by western blotting. We obtained two antibodies against BrC-Z2 and BrC-Z4 using synthetic peptides from zine finger domains to immunize rabbits. Antibodies are specific in the recombinant proteins, but unfortunately, we cannot detect the specific signal in the silk gland. We think antibodies against BrC-Z2 and BrC-Z4 by synthetic peptides might be some problem and plan to prepare antibodies using recombinant proteins. Therefore, only semi RT-PCR and qRT-PCR were used to investigate the expression patterns of relative genes.

Point 2: To analyze the gene function, usually gene deletion or knock-down is required. Is it possible to delete or knock down the expression of BrC-Z by RNA interference?

Response 2: Thanks for your suggestion. Fibroin genes are expressed in the inter-molting stage and are repressed in the molting stage, but BrC-Zs are hardly expressed in the inter-molting stage in the silk gland. We guess BrC might be a negative regulatory factor of fibroin genes. Thus, we chose the ectopically transgenic overexpression in the PSG driven by fib-h promoter for exploring the potential relationship between BrC and fibroin genes. Knocking out or knocking down of BrC is not suitable to the negative regulation of fibroin genes in the inter-molting stages. Moreover, BrC is a key factor of the 20E signal pathway, which play very important roles in the grow development and metamorphosis.

Point 3: How about the mechanism of this regulation? Could give the signal pathway?

Response 3: Thanks for your valuable suggestion, we provide a graphic model to illustrate a potential regulatory mechanism.

Point 4: Poor English writing. Lots of error in grammar and writing. It is necessary to improve it by native English writers.

 Response 4: Following your suggestion, we have invited an experienced English editor from Editage, a professional English Editing company, to help us revise English grammatical errors.

Point 5: Figure 1. The ribosomal protein L3 (RPL3) gene of the silkworm was used as the reference gene. But, there are significant decrease of RPL3 in Wandering 2d compared with other period in the figure. Besides, it is better to provide the ratio of genes of interest to the control RPL3 because the luminance of RPL3 in 2nd Day of wandering seems to differ from other stages.

Response 5: Thanks for pointing this problem. Because BrC-Zs are hardly expressed in from the V3rdD to the pupae stages, so we did not normalize the RPL3 in Wandering 2d.

Point 6: Figure 2A. The positive transgenic phenotype was observed in the pupae. Could this positive phenotype be maintained in 4th instar, 5th larva? Please offer the relative expression level of BrC-Z2 between transgenic overexpression line and wild-type during the designed time points. *Figure 4. Comparing with BrC-Z2-OE line treatment with JH and DMSO, the author should also supplement data with wild-type treatment with JH or DMSO respectively.

Response 6: Thanks for your questions. The red fluorescence of pupae in Fig2A was used to screen positive transgenic individuals, which was driven by the eye-specific 3xP3 promoter. As you said, we initially investigated the expression pattern of BmBrC-Z2 by RT-PCR from V1stD to from V7thD and showed a continuous expression pattern. For qRT-PCR, we just selected a time point (V5thD) to investigate the expression levels of related genes in Fig 3. There were no significant change of cocoon layer ratio (cocoon weight/total weight) of the WT after JH treatment (Fig. 4B). In our previous study (Zhao et al., JBC, 2015), JH treatment could significantly increase expression levels of genes, such as BmKr-h1, Bmdimm, and Fib-h. Here, we did not investigate the expression patterns of these genes in the WT when treated with JH and only cited the reference in discussion section.

Point 7: “ 3.3”, appears twice.

Response 7: we are very sorry for the error.

Point 8: Line130: The expression level started to recover at V1stD instead of V3rdD according to figure1.

Response 8: Thank you point this writing error.

Point 9: Line173: It is inadequate to determine the hypothesis mentioned in this line without monitoring JH concentration in transgenic lines.

Response 9: Thank you for pointing the problem. JH treatment can really rescue the phenotype of the transgenic line. And JH also promote the expression levels of fibroin genes and regulatory factors of JH signal pathway, not only in this study, but also in a previous study (Zhao, et al., JBC, 2015). It is a pity that there is no mature method how to monitor JH concentration in the silk gland. We also plan to test the JH concentration in silk gland by HPLC in the future.

Point 10: Line29: “resulted to” should be “resulted in”. “active the target genes in insects” ?

Response 10: Thanks, we have revised it.

Reviewer 3 Report

In this manuscript, the authors revealed the relations between fibroin genes, Broad-Complex (BrC) and JT signal pathway in Bombyx mori. They investigated kinetics of expression of BmBrC-Zs and fibroin genes, suggesting that BmBrC-Zs function as negative regulators. Consequently, they showed that thinner and lighter color cocoons were brought about by overexpression of BmBrC-Z2 in PSG, and actual lower expression of fibroin genes and JH related transcription factors (BmDimm and BmKr-h1). The results suggested that BmBrC-Z2 actually involved in cocoon forming and coloring by negatively regulating fibroin genes and there is relationships between BmBrC and JH signaling pathway. The authors conducted experiments using  BmBrC-Z2 overexpression lines and JH analogue Methoprene (JHA). The experiments revealed that phenotypes of cocoons and gene expressions of Fibroin genes and JH related transcription factors were shifted to these of normal ones. The authors showed the relations between he relations between fibroin genes, Broad-Complex (BrC) and JT signal pathway in Bombyx mori.

It is a important paper for insect community, because the authors shed new insights of relations between  JH signaling and cocoon forming. In my opinion, this paper is worth to publishing in “Insects”.  However, there are several points to be addressed for improving the manuscript.

  1. The authors used BmBrC-Z2 overexpression lines in this manuscript. Why they did not use BmBrC-Z4? 
  2. In introduction, the authors listed BmSage or the other transcription factors (lines 49-51). They must investigated expression levels of them as BmDimm and BmKr-h1
  3. The authors discussed relationships between BmBrC, Fibroin genes and JH signaling. Scheme figure showing the relationships should be added for readers to easily understand their work.

Author Response

In this manuscript, the authors revealed the relations between fibroin genes, Broad-Complex (BrC) and JT signal pathway in Bombyx mori. They investigated kinetics of expression of BmBrC-Zs and fibroin genes, suggesting that BmBrC-Zs function as negative regulators. Consequently, they showed that thinner and lighter color cocoons were brought about by overexpression of BmBrC-Z2 in PSG, and actual lower expression of fibroin genes and JH related transcription factors (BmDimm and BmKr-h1). The results suggested that BmBrC-Z2 actually involved in cocoon forming and coloring by negatively regulating fibroin genes and there is relationships between BmBrC and JH signaling pathway. The authors conducted experiments using BmBrC-Z2 overexpression lines and JH analogue Methoprene (JHA). The experiments revealed that phenotypes of cocoons and gene expressions of Fibroin genes and JH related transcription factors were shifted to these of normal ones. The authors showed the relations between he relations between fibroin genes, Broad-Complex (BrC) and JT signal pathway in Bombyx mori.

It is an important paper for insect community, because the authors shed new insights of relations between JH signaling and cocoon forming. In my opinion, this paper is worth to publishing in “Insects”.  However, there are several points to be addressed for improving the manuscript.

Point 1: The authors used BmBrC-Z2 overexpression lines in this manuscript. Why they did not use BmBrC-Z4? 

Response 1: Thank you very much for pointing this problem. In Fig. 1, we found that the expression patterns of BmBrc-Z2 and BmBrc-Z4 were similar by RT-PCR analysis. By qRT-PCR analysis, we found the relative expression level of BmBrc-Z2 was higher than that of BmBrc-Z4. The result was shown as Fig. 1B. Thus, we first chose BmBrc-Z2 to ectopically overexpress in the PSG.

Point 2: In introduction, the authors listed BmSage or the other transcription factors (lines 49-51). They must investigated expression levels of them as BmDimm and BmKr-h1. 

Response 2: As you said, actually, we have investigated the expression patterns of SGF1 and BmSage. There were no significant changes of them between the transgenic line and the wild type. So, we did not show those results in the manuscript.

Point 3: The authors discussed relationships between BmBrC, Fibroin genes and JH signaling. Scheme figure showing the relationships should be added for readers to easily understand their work.

Response 3: Thanks for your valuable suggestion, we provide a graphic model to illustrate a potential regulatory mechanism.

Reviewer 4 Report

Silk protein biosynthesis in the silk gland cells of silkworms is one of the most attractive topics for silkworm research. The polypoid properties (the number of cells keeps stable, but the DNA in the cells can be amplified millions of times) of the silk gland cells are not only valuable to silk production, but also to studies in mammalian cells. However, there are still many unknown mechanisms needs to be further investigated. Based on the gene expression research, the authors found that two BmBrC isoforms exhibited contradictory expression patterns with fibroin genes (FibH, FibL, and P25) in the posterior silk gland (PSG), suggesting that BmBrC might negatively regulate fibroin genes. Then, the authors constructed transgenic lines whereBmBrC-Z2 was overexpressed in the posterior silk gland (PSG). The authors found that the silk protein genes in the transgenic line were decreased to almost half of that in the wild type and the silk yield was also decreased significantly. Moreover, the expression levels of regulatory factors (BmKr-h1 and BmDimm) response to JH signal were inhibited significantly. Therefore, the authors concluded that BmBrC might inhibit fibroin genes by repressing the JH signal pathway.

Major comments:

First, in figure 1, both BmBrc-Z2 and BmBrc-Z4 showed similar expression patterns, however, authors only provided the data for BmBrc-Z2 in this manuscript. It is necessary to add the data for BmBrc-Z4 or to explain the reason why BmBrc-Z2 alone was chosen to test the regulatory effects on silk production. It is highly recommended that the second gene is included in this work to give a whole story.

Second, it is better to provide direct evidence that BmBrCs inhibit the JH signal pathway by including expression analysis of genes in the 20-E and JH pathway in the ectopic over-expressed BmBrC-Z2 transgenic line.

Minor comments:

  1. The expression of BmBrC-Z2 in transgenic lines with ectopic overexpression of BmBrC-Z2 should be tested at the time course like the genes in figure 1, from V1stD to pupae.
  2. Did the authors find any other phenotypes such as the pupae weight, cocoon weight and development time in the transgenic lines?
  3. The complex regulatory mechanisms of the inhibition of silk protein genes by BmBrC can be better presented with a graphic explanatory model.
  4. The English vocabulary, tense, and grammar need to be revised thoroughly.

Author Response

Silk protein biosynthesis in the silk gland cells of silkworms is one of the most attractive topics for silkworm research. The polypoid properties (the number of cells keeps stable, but the DNA in the cells can be amplified millions of times) of the silk gland cells are not only valuable to silk production, but also to studies in mammalian cells. However, there are still many unknown mechanisms needs to be further investigated. Based on the gene expression research, the authors found that two BmBrC isoforms exhibited contradictory expression patterns with fibroin genes (FibH, FibL, and P25) in the posterior silk gland (PSG), suggesting that BmBrC might negatively regulate fibroin genes. Then, the authors constructed transgenic lines whereBmBrC-Z2 was overexpressed in the posterior silk gland (PSG). The authors found that the silk protein genes in the transgenic line were decreased to almost half of that in the wild type and the silk yield was also decreased significantly. Moreover, the expression levels of regulatory factors (BmKr-h1 and BmDimm) response to JH signal were inhibited significantly. Therefore, the authors concluded that BmBrC might inhibit fibroin genes by repressing the JH signal pathway.

Major comments:

Point 1: First, in figure 1, both BmBrc-Z2 and BmBrc-Z4 showed similar expression patterns, however, authors only provided the data for BmBrc-Z2 in this manuscript. It is necessary to add the data for BmBrc-Z4 or to explain the reason why BmBrc-Z2 alone was chosen to test the regulatory effects on silk production. It is highly recommended that the second gene is included in this work to give a whole story.

Response 1: Thank you very much for pointing this problem. In Fig. 1, we found that the expression patterns of BmBrc-Z2 and BmBrc-Z4 were similar by RT-PCR analysis. By qRT-PCR analysis, we found the relative expression level of BmBrc-Z2 was higher than that of BmBrc-Z4. The result was shown as Fig. 1B. Thus, we first chose BmBrc-Z2 to ectopically overexpress in the PSG.

Point 2: Second, it is better to provide direct evidence that BmBrCs inhibit the JH signal pathway by including expression analysis of genes in the 20-E and JH pathway in the ectopic over-expressed BmBrC-Z2 transgenic line.

Response 2: Thanks for your good suggestion. We think that BmBrC might inhibit the expression of Kr-h1 in the JH signal pathway. Work on their regulatory mechanism is under way.

Minor comments:

Point 3: The expression of BmBrC-Z2 in transgenic lines with ectopic overexpression of BmBrC-Z2 should be tested at the time course like the genes in figure 1, from V1stD to pupae.

Response 3: Thank you pointing the problem. As you said, we initially investigated the expression pattern of BmBrC-Z2 by RT-PCR from V1stD to from V7thD and showed a continuous expression pattern. For qRT-PCR, we just selected a time point to investigate the expression levels of related genes.

Point 4: Did the authors find any other phenotypes such as the pupae weight, cocoon weight and development time in the transgenic lines?

Response 4: Thanks for your question. There is no change in the development period of the transgenic lines comparing with the wild-type strains. We found that the main influence of ectopic over-expressed BmBrC-Z2 is the economic character. The pupae weight has no changed significantly and the cocoon layer weight decreases significantly, which result in a significant reduction in the cocoon layer ratio (cocoon layer weight/total weight).

Point 5: The complex regulatory mechanisms of the inhibition of silk protein genes by BmBrC can be better presented with a graphic explanatory model.

Response 5: Thanks for your valuable suggestion, we provide a graphic model to illustrate a potential regulatory mechanism.

Point 6: The English vocabulary, tense, and grammar need to be revised thoroughly.

Response 6: Following your suggestion, we have invited an experienced English editor from Editage, a professional English Editing company, to help us revise English grammatical errors. 

Round 2

Reviewer 1 Report

The revised manuscript has been improved greatly.

Reviewer 3 Report

The authors addressed properly reviewer's comments. So I recommend the manuscript is accepted.